# Surface Properties of CVD-Grown Graphene Transferred by Wet and Dry Transfer Processes

**DOI:** 10.3390/s22103944

**Published:** 2022-05-23

**Authors:** Min-Ah Yoon, Chan Kim, Jae-Hyun Kim, Hak-Joo Lee, Kwang-Seop Kim

**Affiliations:** 1Division of Mechanical Engineering, University of Science & Technology (UST), 217 Gajeong-ro, Yuseong-gu, Daejeon 34113, Korea; mayoon@ust.ac.kr (M.-A.Y.); jaehkim@kimm.re.kr (J.-H.K.); 2Nano-Convergence Mechanical Systems Research Division, Korea Institute of Machinery & Materials (KIMM), 156 Gajeongbuk-ro, Yuseong-gu, Daejeon 34103, Korea; chankim@kimm.re.kr; 3Center for Advanced Meta-Materials (CAMM), 156 Gajeongbuk-ro, Yuseong-gu, Daejeon 34103, Korea; hjlee@kimm.re.kr

**Keywords:** surface properties, contact angles, CVD-grown graphene, wet transfer method, dry transfer method

## Abstract

Graphene, an atomically thin material, has unique electrical, mechanical, and optical properties that can enhance the performance of thin film-based flexible and transparent devices, including gas sensors. Graphene synthesized on a metallic catalyst must first be transferred onto a target substrate using wet or dry transfer processes; however, the graphene surface is susceptible to chemical modification and mechanical damage during the transfer. Defects on the graphene surface deteriorate its excellent intrinsic properties, thus reducing device performance. In this study, the surface properties of transferred graphene were investigated according to the transfer method (wet vs. dry) and characterized using atomic force microscopy, Raman spectroscopy, and contact angle measurements. After the wet transfer process, the surface properties of graphene exhibited tendencies similar to the poly(methyl methacrylate) residue remaining after solvent etching. The dry-transferred graphene revealed a surface closer to that of pristine graphene, regardless of substrates. These results provide insight into the utilization of wet and dry transfer processes for various graphene applications.

## 1. Introduction

Most gas sensors operate in a resistive mode with a simple mechanism, in which the electrical resistance of the sensing material changes after a chemical reaction with an analyte [1]. The actual sensing materials thus play a key role in determining sensor performance. Gas sensors mainly utilize metal oxide films as the sensing material to precisely detect gas leakage at temperatures above a few hundred degrees Celsius [2,3]. The sensor’s problems are that the high operating temperature of gas sensors with metal oxide films can trigger an explosion of gases, such as hydrogen and oxygen. The limited flexibility of the sensors due to metal oxide films makes curved attachments on gas pipes and vessels unsuitable [4]. For this reason, graphene and graphene composites have been suggested as a sensing material for gas sensors due to their excellent electrical and mechanical properties, as well as their flexibility [5,6,7,8].

Chemical vapor deposition (CVD)-grown graphene has advantages for fabricating large-area and high-performance gas sensors rather than mechanical and liquid exfoliation graphene [9]. CVD graphene is generally synthesized on a metallic catalyst; thus, the graphene should be transferred onto a target substrate to fabricate gas sensors [10,11]. Although various transfer processes of graphene, such as wet, dry, mechanical, electrochemical, and polymer-free transfer, have been suggested, wet and dry transfer methods have been widely utilized because the methods can be simply performed using polymeric films as a medium in the transfer process and produce high-quality graphene [12,13,14]. Despite their common use, chemical modification and mechanical damages are still caused to graphene surfaces during wet and dry transfer processes [5,15,16,17,18]. Interestingly, it has been reported that the damages to transferred graphene rather improve the sensitivity of gas sensors because linear wrinkles and cracks in graphene restrict the path of electrons, thus increasing the electrical resistance of graphene [5]. In addition, polymeric residue on the graphene surface of gas sensors acts as a high-sensitivity functionalization layer for analyte adsorption [15]. However, this type of damage reduces the charge carrier mobility of graphene, thereby degrading the performance of the gas sensor; also, the residue increases the binding energy with gas molecules, resulting in gas sensor recovery issues. Recently, it has been reported that the charge carrier mobility and reversibility of graphene sensors can be enhanced by screening for unintentional substrate-induced p-doping of the graphene using a hydrophobic polymer brush layer on a target substrate [19]; it was also suggested that the gas adsorption rate of graphene depends on underlying substrates due to differences in the surface energy of the substrates. The surface energy of CVD-grown graphene on copper foil immediately after synthesis is about 10~30% higher than that of graphene after 24 h of air exposure because airborne hydrocarbon is adsorbed onto graphene [20,21]. Moreover, the surface energy of single-layer graphene is changed by underlying substrates [22]. Thus, high-performance graphene gas sensors require graphene with low and stable surface energy regardless of underlying substrates, as well as minimal surface damage to the graphene during the transfer process.

The wet transfer process of graphene is performed by scooping a floating graphene/polymer film onto a target substrate [23]. After scooping, the polymer film supporting the graphene is removed with a solvent to prevent mechanical damage. The dry transfer process is performed based on the differences in adhesion between the interfaces of the graphene/polymer film and graphene/substrate. A transfer film with a pressure-sensitive adhesive (PSA) layer is used to support the graphene during the dry transfer process [14]. Once the graphene has adhered to the target substrate, the transfer film is peeled away from the graphene. Since the two transfer methods have distinct differences in transfer environment and mechanism, the surface properties of graphene can vary due to the chemical modification and mechanical damages in graphene regarding the transfer methods. Therefore, to produce high-performance graphene-based gas sensors, the surface properties of the transferred graphene depending on the transfer methods should be understood.

Here, we investigated the surface and electrical properties of graphene transferred onto a target substrate using wet and dry transfer processes. The wet-transferred graphene showed a lower sheet resistance of 207 Ω/Sq and less damage compared with the graphene transferred using the dry transfer process; however, the wet-transferred graphene was covered by a 1.5-nm-thick poly(methyl methacrylate) (PMMA) residue. Moreover, the water contact angle (WCA) of the wet-transferred graphene was comparable to that of PMMA, whereas the WCA of dry-transferred graphene was close to that of graphite (HOPG). In addition, the wet-transferred graphene exhibited a higher surface energy than that of dry-transferred graphene due to the PMMA residue, and the surface energy of the wet-transferred graphene differed according to the substrate. In contrast, the surface energy of dry-transferred graphene was analogous to that of HOPG, regardless of the substrate. Taken together, these results indicate that the surface and electrical properties of graphene, especially the surface energy, depend on the transfer method. Finally, we propose that the dry transfer process has many advantages for producing high-quality graphene-based gas sensors in an industrial aspect.

## 2. Materials and Methods

### 2.1. Sample Preparation

Monolayer graphene (Gr) was synthesized on 35 μm-thick Cu foil (JX Nippon Mining and Metals Corp., Tokyo, Japan) using a thermal CVD process [24]. Polydimethylsiloxane (PDMS, Sylgard 184; Dow Corning, Midland, MI, USA) and polyethylene terephthalate (PET) film were used as the compliant layer and supporting film in the transfer film (TF), respectively. To enhance the adhesion between PDMS and PET, plasma surface treatment (CUTE Plasma System; Femto Science, Gyeonggi-do, South Korea) was performed on PET before applying the PDMS coating. The PDMS was prepared by mixing a liquid prepolymer (Sylgard 184A; Dow Corning) and curing agent (Sylgard 184B; Dow Corning). The mixture was spin-coated as 100-μm-thick layers onto the PET film and baked for 12 h at 60 °C [14].

### 2.2. Dry Transfer Process

Figure 1a shows a schematic diagram of the dry transfer process for monolayer CVD-grown Gr on Cu foil. The TF was laminated onto Gr grown on Cu foil using a home-built roll-to-plate (R2P) transfer machine [25,26]. The Cu foil was etched with 0.1 M ammonium persulfate solution (APS; Sigma-Aldrich, St. Louis, MO, USA) [24]. After etching, the TF/Gr film was laminated onto the target substrate under a contact load of 2 N/mm and lamination speed of 0.5 mm/s using the R2P transfer machine. For the roll-based large-area dry transfer, the TF was peeled off with the R2P system under precise active load control, using a contact load and peeling-off velocity of 0.2 N/mm and 0.1 mm/s, respectively. The radius of the roller in the R2P system was 75 mm. (Appendix A).

### 2.3. Wet Transfer Process

Figure 1b shows a schematic diagram of the wet transfer process for monolayer CVD-grown Gr on Cu foil. PMMA (Microchem Laboratory, Round Rock, TX, USA) was spin-coated onto Gr grown on Cu foil and baked for 1 min at 80 °C [23]. The PMMA/Gr/Cu foil sample was then floated on a 0.1 M APS solution to etch away the Cu foil. After etching, the PMMA/Gr was floated on deionized (DI) water to remove any etchant impurities and then scooped onto the target substrate. Then, the PMMA on Gr was removed by placing the PMMA/Gr/substrate in an acetone bath for 30 min.

### 2.4. Characterizations

Raman spectra were obtained using a Raman spectrometer with a 514-nm laser as the excitation source (inVia Raman Microscope; Renishaw, Wotton-Under-Edge, UK). The beam size of the laser was 2 μm and a 50× objective lens was used. Images of the transferred Gr were obtained using field emission scanning electron microscopy (FE-SEM; JSM-7610 FPlus; JEOL, Ltd., Tokyo, Japan) operating at less than 1 kV to suppress charging. The topography of the samples was examined using atomic force microscopy (AFM; 5300E; Hitachi, Ltd., Tokyo, Japan). An Si probe (SI-DF3; Hitachi, Ltd.) with a stiffness of 1.6 N/m and resonance frequency of 27 kHz was used to measure the topography over an area of 5 × 5 μm^2^ at a scan rate of 0.5 Hz. The sheet resistance of the Gr transferred onto the SiO_2_/Si substrate was measured using a four-point probe nanovoltmeter (Model 6221; Keithley Instruments, Cleveland, OH, USA). The contact angle was measured using a drop shape analyzer (DAS 100; Krüss GmbH, Hamburg, Germany). As polar and dispersive solutions, 3 μL droplets of DI water, formamide, and diiodomethane were dropped onto different areas of the substrate at least three times. All contact angle data were obtained within 5 h after the transfer process.

## 3. Results and Discussion

The surface of CVD-grown Gr transferred onto an SiO_2_ substrate was observed using an optical microscope and SEM, as shown in Figure 2a,b. After the Gr transfer, we observed little mechanical damage; specifically, few cracks and wrinkles were present, regardless of the transfer method (Figure 2a). However, the wet-transferred Gr showed a residue (Figure 2b). Raman spectroscopy was used to examine the chemical modifications of the transferred Gr onto SiO_2_ (Figure 2c). The Raman spectra displayed G and 2D peaks of intrinsic Gr, regardless of the transfer method. The 2D-band/G-band ratio, I_2D_/I_G_, of dry-transferred Gr was about 1.9, which is close to the value of 2 for pristine monolayer Gr. The Raman spectra of wet-transferred Gr, I_2D_/I_G_ was 1.1, and the G and 2D peaks had shifted slightly to ω_G_ ~ 1596 cm^−1^ and ω_2D_ ~ 2694 cm^−1^ from its intrinsic G- and 2D-bands (ω_G_ ~ 1580 cm^−1^ and ω_2D_ ~ 2680 cm^−1^) [27,28,29]. The Raman spectra of dry-transferred Gr on SiO_2_ appeared to be pristine Gr; however, the wet-transferred Gr showed a chemically modified surface (Appendix A). The chemical modifications were likely due to the PMMA residue on the surface of Gr. If PMMA is not removed completely from the Gr surface, the residue effectively acts as a p-type dopant, which degrades the charge carrier mobility of graphene [19].

The sheet resistance of Gr was measured after wet (Figure 2d) and dry (Figure 2e) transfer. The sheet resistance of wet-transferred Gr was uniform in the area where it was measured, and the average sheet resistance was 450 ± 106 Ω/Sq (Figure 2d). Compared with the wet transfer process, the dry-transferred Gr showed poor uniformity in terms of its sheet resistance; the average sheet resistance was 657 ± 130 Ω/Sq, which is about 1.5-fold that of the wet-transferred Gr (Figure 2e). In the dry transfer process, Gr was adhered to the TF by van der Waals (vdW) interactions (a relatively weak interaction force). Thus, fine cracks of Gr on the TF were easily generated during the etching process of Cu and the roll-based lamination process [14]. On the contrary, PMMA prevented tear and crack formation in Gr due to the strong chemical bonds formed with PMMA; however, this strong bonding is also responsible for the difficulty in removing PMMA from the Gr surface. 

AFM was used to further examine the surface properties of Gr transferred onto SiO_2_, as shown in Figure 3. Figure 3a shows a schematic diagram of the scanning procedure for analyzing the surface morphology of the transferred Gr, which might include polymeric residue, cracks, and wrinkles. First, the surface morphology of samples was obtained over scanning areas of 5 × 5 μm^2^ (Figure 3b,d) and 3 × 3 μm^2^ (Figure 3c) in non-contact mode. Second, to confirm the presence of a polymeric residue on the Gr surface, the surfaces of samples were scanned using a cantilever under a 2-nN load in contact mode, thus effectively scraping over the samples, as shown by the 2 × 2 μm^2^ black dotted area in Figure 3b–d and the 1 × 1 μm^2^ area in Figure 3c. Finally, the scanned area in the first step was rescanned to verify the changes in surface morphology in non-contact mode. Figure 3b–d shows topography images of the surfaces of Gr/Cu and the wet- and dry-transferred Gr/SiO_2_ obtained in the final step of our AFM analysis. The topography images of Gr synthesized on Cu show step patterns with a root-mean-square (RMS) roughness of 7.74 nm, which were formed by reconstructing the surface of Cu during the synthesis of Gr at high temperatures (Figure 3b). The surface of Gr/Cu was not significantly changed after the scraping step and application of the AFM probe in contact mode. After the wet transfer process, however, the surface of Gr/SiO_2_ had an RMS roughness of 6.4 nm, and there was visible residue accumulation at the edges of the scraped area on Gr/SiO_2_ (Figure 3c). Compared with the surface of wet-transferred Gr/SiO_2_ covered with the residue, the surface of the dry-transferred Gr/SiO_2_ was slightly flattened, with an RMS roughness of 5.13 nm, although trapped bubbles between Gr and SiO_2_ were observed; in addition, there was no evidence of residue on the edge of the scraped area (Figure 3d). To measure the thickness of the polymeric residue on Gr/SiO_2_, we examined the line profiles of the topography images (white dotted lines in Figure 3e–g). Figure 3e shows similar surface profiles of Gr/Cu between scraped and non-scraped areas. However, the line profile of wet-transferred Gr/SiO_2_ displayed a distinct step of 1.5 nm thickness at the edge of the scraped area, corresponding to the thickness of the PMMA residue on the surface of Gr after the wet transfer process (Figure 3f). In contrast, in the line profiles of dry-transferred Gr/SiO_2_, the scraped and non-scraped areas could not be distinguished (Figure 3g). Thus, the results shown in Figure 3 confirm that the PMMA residue in the Gr after the wet transfer process was not locally present but instead covered the entire area, whereas there was no polymeric residue on dry-transferred Gr.

To investigate the effects of the transfer methods and substrates on the wettability of Gr, the contact angles of three solutions, i.e., DI water (polar), formamide (polar and dispersive), and diiodomethane (dispersive), were measured on the surfaces of a bare substrate (black bar), wet-transferred Gr/substrates (red bar), and dry-transferred Gr/substrates (blue bar) (Figure 4). In our case, all contact angles (CAs) of samples were measured after 48 h of graphene synthesis, and CAs were not significantly changed in each sample for 30 min of the measurement. Therefore, we considered that airborne hydrocarbons were adsorbed onto the surface of graphene before CA measurement [20,21]. Figure 4a shows the water contact angle (WCA) of the samples. The WCAs of bare SiO_2_, p-PET, and h-PET were 44°, 72°, and 42°, respectively. The WCAs of SiO_2_ and h-PET were 28–30° lower than that of p-PET, because the oxide layer on the substrates formed hydrogen bonds with the water molecules. After the wet transfer process, the WCAs of the Gr/substrate were 64.2° (SiO_2_), 76.4° (p-PET), and 79.8° (h-PET). Thus, the average contact angle of wet-transferred Gr on the substrates was 73.5 ± 8° compared with 76° for PMMA. The WCAs of the dry-transferred Gr/substrate increased to 86.2° (SiO_2_), 90.6° (p-PET), and 88.4° (h-PET), with an average contact angle of 88.4 ± 2°; this was comparable with 96° for HOPG. The formamide contact angles (FCAs) of the bare substrates were 14° (SiO_2_), 55.8° (p-PET), and 41° (h-PET) (Figure 4b). Because formamide has both dispersive and polar energy properties, it has a stronger interaction with SiO_2_ and h-PET compared to p-PET [30]. The FCAs of the wet-transferred Gr/substrate were 57.2° (SiO_2_), 62.5° (p-PET), and 63.8° (h-PET), and the average contact angle was 61.2 ± 3°, similar to the value of 61° observed for PMMA. Dry-transferred Gr on SiO_2_, p-PET, and h-PET exhibited FCAs of 59.7°, 68.7°, and 68.9°, respectively. The average FCA value of Gr/substrate was 65.8 ± 5°, which was close to the 69° of HOPG. The diiodomethane contact angles (DCAs) of the bare substrates were 50.5° (SiO_2_), 27.3° (p-PET), and 41° (h-PET) (Figure 4c). The DCA results showed that the DCA of p-PET was about 13.7~23.2° lower than that of SiO_2_ and h-PET. Because the surface of p-PET is dispersive, diiodomethane tends to interact strongly with p-PET. Thus, the DCA of the dry-transferred Gr/substrate, which had no residue on its surface, was expected to be lower than that of the wet-transferred Gr/substrate. However, DCA on the Gr/substrate was nearly constant at 43 ± 2°, regardless of the transfer method or substrate. In addition, the DCA of the Gr/substrate was about 8° higher than that of HOPG and 4° lower than that of PMMA. 

The surface energy of the Gr/substrate was calculated using the Young–Dupre [31] and Owen–Wendt [32] theories (Figure 5), based on the contact angle measurements obtained for DI water, formamide, and diiodomethane of the transferred Gr/substrate. Figure 5a shows the surface energies of Gr transferred onto the substrates. The surface energy of the wet-transferred Gr/substrate varied from 35.2 to 42.3 mJ/m^2^, depending on the substrate, which was 10–25% lower than that of the bare substrates. Regardless of the substrates, the surface energies of the dry-transferred Gr/substrate varied slightly, between 35.5 and 37.2 mJ/m^2^, which was 10–35% lower than that of the bare substrates. Figure 5b shows dispersive and polar energy terms in the surface energy equation of wet-transferred Gr on the substrates. The dispersive energy of the wet-transferred Gr/substrate was constant over the range of 27.6–30.5 mJ/m^2^; however, the polar energy of wet-transferred Gr/SiO_2_ was 14.8 mJ/m^2^, about two-fold that of wet-transferred Gr on p-PET and h-PET. The average energy values, regardless of the substrate, were 29 ± 1.4 mJ/m^2^ (dispersive) and 9.8 ± 4.4 mJ/m^2^ (polar), which were comparable to PMMA (30.5 mJ/m^2^ for the dispersive energy and 7.9 mJ/m^2^ for the polar energy). In contrast, the energy terms of the dry-transferred Gr/substrate did not change significantly by substrate (Figure 5c). The average dispersive energy of the dry-transferred Gr/substrate was 33.1 ± 1.1 mJ/m^2^, which was 11.4 times higher than its average polar energy, and the averaged energies were analogous to those of HOPG. These results were due to the bubbles that existed in the interface of the dry-transferred Gr/substrate, making a relatively large gap between the dry-transferred Gr and underlying substrates. Taken together, the results shown in Figure 2, Figure 3, Figure 4 and Figure 5 indicate that the PMMA residue on the wet-transferred Gr surface clearly modified the surface properties of Gr; however, the dry-transferred Gr remained relatively pristine. It has been reported that contamination and ambient pollution, such as CO_2_, O_2_, and H_2_O, act as interference factors that degrade the performance of Gr [19,33,34], leading to a decline in sensor performance or even impaired development. In addition, the contact angles of Gr can increase as the exposure time of Gr to the atmosphere increases the adsorption of airborne hydrocarbons [20,33]. Thus, the aging effect on the surface energy of Gr should be considered in the design process of gas sensors. The results of our study demonstrate the many advantages of using the dry transfer process for Gr transfer to minimize polymeric residue, low doping, low surface energy, and productivity issues, compared with the wet transfer process for Gr.

## 4. Conclusions

In this study, we investigated the surface and electrical properties of both wet- and dry-transferred Gr on various substrates. Wet- and dry-transferred Gr showed fewer mechanical damages such as cracks and wrinkles, but PMMA residues and bubbles were observed, respectively. Gr and the PMMA layer form strong bonds in the wet transfer process, which make it difficult to remove PMMA from the Gr surface, as evidenced by the remaining PMMA residue. Raman spectra of wet-transferred Gr on SiO_2_ showed a reduction in the I_2D_/I_G_ ratio to 1.1, and the G and 2D peaks were shifted slightly from the intrinsic G and 2D bands. Based on the Raman spectra, the dry-transferred Gr on SiO_2_ appeared to be pristine Gr; however, the wet-transferred Gr showed chemical modification. The sheet resistance of the wet-transferred Gr/substrate was 450 Ω/Sq, which was 30% lower than that of the dry-transferred Gr/substrate. We found that the wet-transferred Gr/substrate was covered with a 1.5 nm-thick PMMA layer, and the residue on Gr behaved as a chemical doping layer, which would degrade the carrier mobility of Gr. No polymeric residue was found on the dry-transferred Gr/substrate; however, trapped bubbles between Gr and the substrate were observed in the topography of AFM images. The surface energy of the Gr/substrate, regardless of the transfer method, was reduced by 10–25% compared to the surface energy on bare substrates. The dispersive and polar energy of the wet-transferred Gr/substrate showed a PMMA-like tendency, but that of the dry-transferred Gr/substrate was comparable to that of HOPG. Therefore, we suggest that the dry transfer process for Gr could be appropriate for graphene-based applications including gas sensors since dry-transferred Gr has advantages in the aspects of fewer contaminants and productivity.

## Figures and Tables

**Figure 1 sensors-22-03944-f001:**
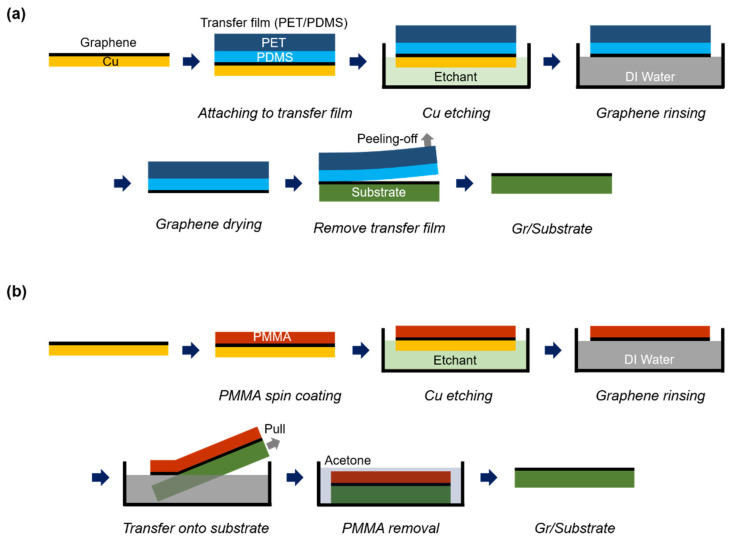
Schematic diagram of the transfer processes of chemical vapor deposition-grown graphene (Gr) on copper foil (Cu): (**a**) dry transfer and (**b**) wet transfer. PDMS: polydimethylsiloxane; PMMA: poly(methyl methacrylate); DI water: deionized water.

**Figure 2 sensors-22-03944-f002:**
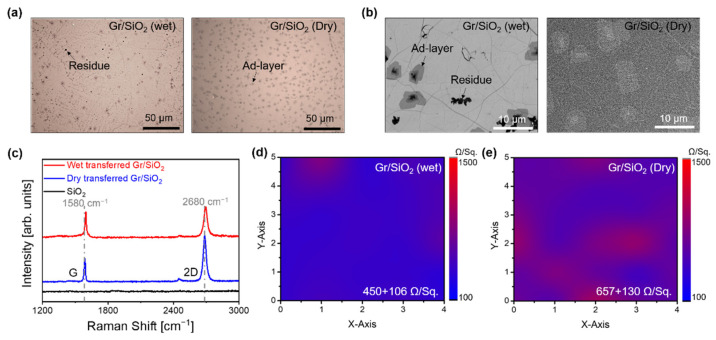
(**a**) Optical microscope and (**b**) scanning electron microscopy images of Gr/SiO_2_ transferred by wet and dry transfer processes. (**c**) Raman spectra of wet- and dry-transferred Gr/SiO_2_ samples. The gray dotted line represents the G and 2D peaks of pristine Gr. (**d**,**e**) Sheet resistance of the Gr/SiO_2_ samples.

**Figure 3 sensors-22-03944-f003:**
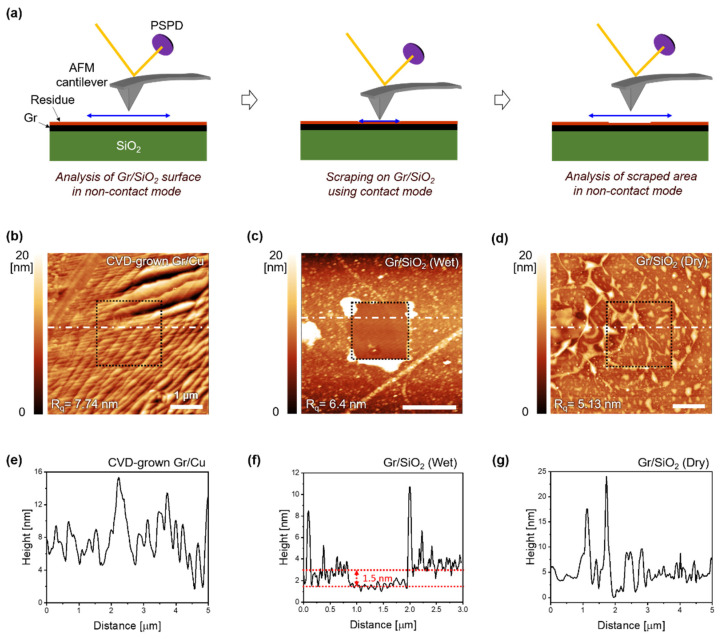
(**a**) Schematic diagram of the residue check procedure using an atomic force microscope. Topography images of (**b**) Gr/Cu, (**c**) wet-transferred Gr/SiO_2_, and (**d**) dry-transferred Gr/SiO_2_. Line profiles on the white dotted line in (**b**–**d**) of (**e**) Gr/Cu, (**f**) wet-transferred Gr/SiO_2_, and (**g**) dry-transferred Gr/SiO_2_. AFM: atomic force microscopy.

**Figure 4 sensors-22-03944-f004:**
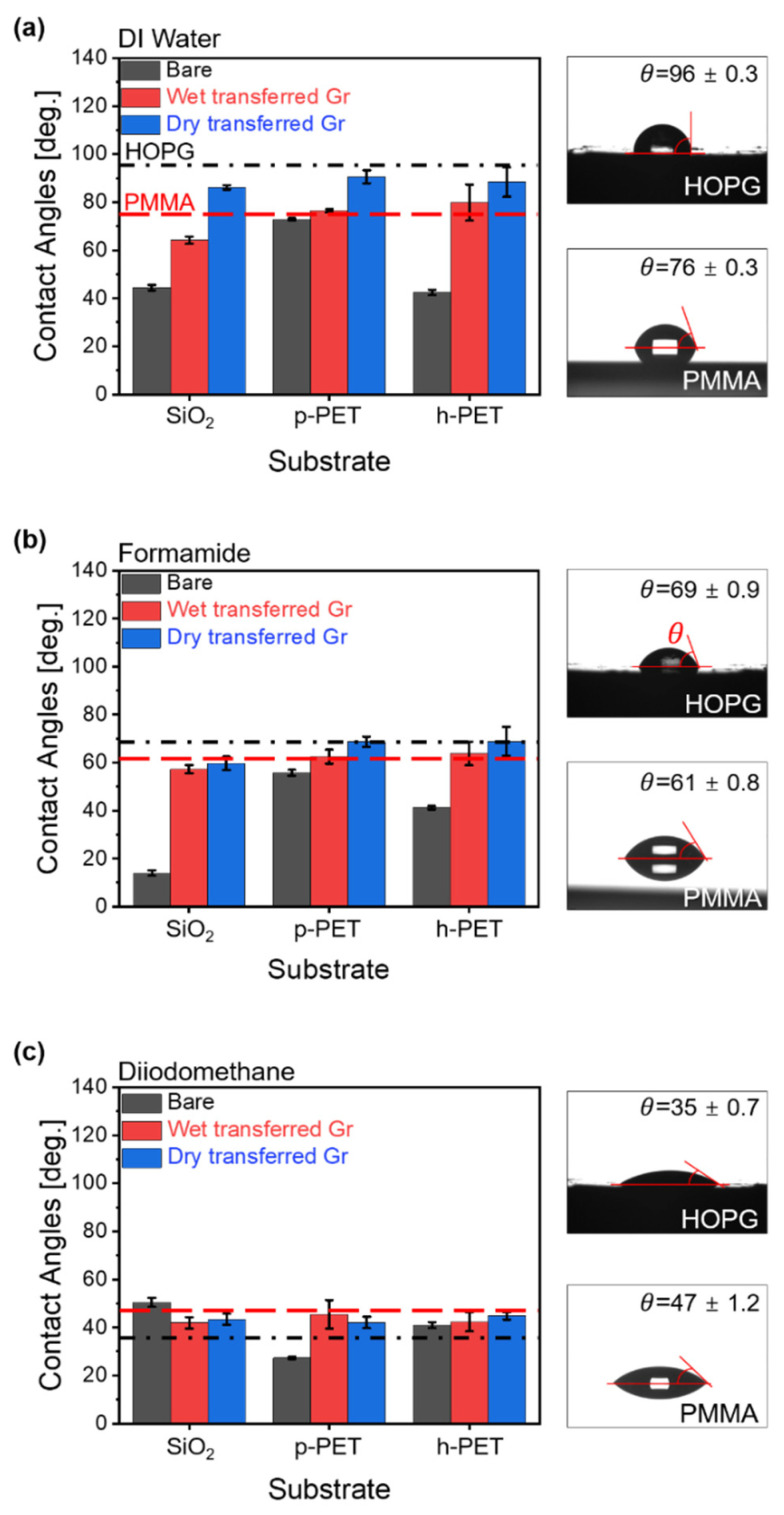
Contact angles of (**a**) deionized (DI) water, (**b**) formamide, and (**c**) diiodomethane on bare substrates, and wet- and dry-transferred Gr/substrate. The black and red dotted lines represent the contact angles of graphite (HOPG) and PMMA, respectively.

**Figure 5 sensors-22-03944-f005:**
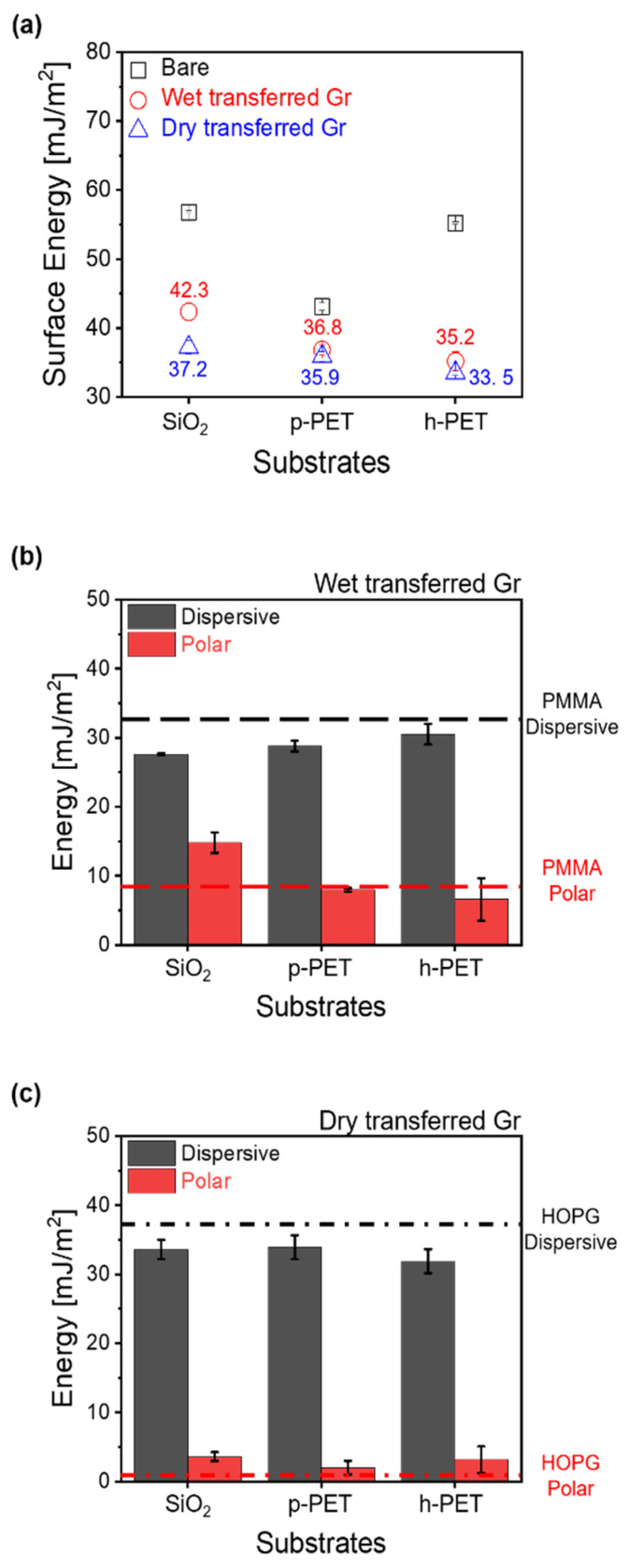
(**a**) Surface energy of bare substrates, and wet- and dry-transferred Gr/substrate. Dispersive and polar energy of (**b**) wet- and (**c**) dry-transferred Gr/substrate. The black and red dotted lines in (**b**,**c**) represent the dispersive and polar energy of PMMA and HOPG, respectively.

## Data Availability

Not applicable.

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
