# Peer review of "Surface Properties of CVD-Grown Graphene Transferred by Wet and Dry Transfer Processes"

_sensors, 2022, doi:10.3390/s22103944_

Round 1

Reviewer 1 Report

In this work, Yoon et al. mainly investigated the surface and electrical properties of Cu-catalyzed CVD-grown graphene which was transferred onto various substrates such as SiO2, h-PET, and p-PET via wet and dry transfer processes. It was observed that the dry transfer process of graphene resulted in low surface energy and high contact angle as compared to the wet-transferred graphene. As far as the graphene growth process and transfer processes are concerned they are already well established and optimized. Therefore, the novelty of the present work is only related to surface energy experiments. Still, there are some important discussions about the surface energy and wetting behavior as well as some important results are missing in this work. Hence, I recommend the manuscript should be revised properly before acceptance. The following points must be addressed:

1) Introduction and discussion sections should include the important information on surface energies of metal-catalyzed CVD-graphene and its wetting behavior because a freshly prepared CVD-grown SLG on a Cu substrate exhibits hydrophilic behavior. However, as the ambient air exposure time increases, its WCA also increases, which is attributed to the adsorption of airborne hydrocarbons on the SLG surface. The initial high surface energy of the freshly prepared SLG continuously decreases as the adsorbed airborne hydrocarbons shield the polar surface sites. Please go through these important articles to strengthen the introduction and discussion sections:

  • "Wetting behaviors and applications of metal-catalyzed CVD grown graphene", J. Mater. Chem. A, 2018, 6, 22437.
  • "Surface-energy engineering of graphene", Langmuir. 2010, 26(6), 3798-802.  
  • "Study on the Surface Energy of Graphene by Contact Angle Measurements", Langmuir. 2014, 30, 28, 8598–8606.

2) If possible please include the side view images of CA to better correlate the results.

3) Why did not you use the other low surface tension organic liquids? What is the rationale behind using formamide and diiodomethane?

4) Could you please elaborate on how did the CA change with time?

5) Sheet resistance should be measured by using the actual analyte then only one can understand the sensing potential of the as-prepared CVD-Gr sample in this case.

Reviewer 2 Report

The manuscript entitled "Surface Properties of CVD-grown Graphene Transferred by Wet and Dry Transfer Processes" compares two transfer techniques of graphene grown through  CVD technique, focusing on the chemical-physical properties of the film transferred with both methods in view of a potential application to sensors.

The work is interesting and falls within the scope of the journal, however it has several weaknesses that need to be addressed to make it suitable for publication.

As a general indication, the whole discussion of the experimental results should be addressed in statistical terms, let me explain better: it is not possible to draw conclusions if the comparison between the two techniques is carried out on the basis of only two samples, 1 for the dry transfer and 1 for wet transfer, especially in consideration of the fact that, in such non-automatic transfer techniques, the operator's hand has a notable impact, which makes the technique not very repeatable. I believe that a statistical analysis is essential to give credibility to the work, therefore at least 20 samples should be prepared with both the one and the other method, in this way the comparison would make sense.

Another weak point of the work is the application to sensors, practically just mentioned in the abstract and in the introduction, after which it is completely lost track, a point to be emphasized with greater incisiveness also in the main text.

Here are some other corrections to be made to the text:

1) Line 34: the authors declare that the MOX sensors are unsuitable for use in the detection of hydrogen because they could trigger a gas explosion, this statement should be supported by a bibliographic reference

2) In lines 50 -51 the authors state “Linear wrinkles and cracks in graphene restrict the path of the electrons, thus increasing the electrical resistance of graphene after analyte adsorption”. It is correct to expect an increase in resistivity in a more defective material as a consequence of a reduction in the electron path, but from this circumstance it is not correct to deduce that the electrical resistivity increases after exposure to the analytes. Exposure to an analyte can manifest itself as both an increase and a decrease in resistivity depending on the interaction with the sensitive film. In the example of graphene, in general, the “as grown” material exhibits a p-type doping and its interaction with electron-donor analytes (for example NH3) is expressed in an increase in resistivity by virtue of the decrease in free carriers. The opposite phenomenon is evident in the interaction with electron-acceptor analytes (for example NO2).

3) Line 106 the side of the TF that is pressed on the graphene must be specified

4) Line 149-150 the statement "The 2D-band / G-band ratio, I2D / IG, of dry-transferred Gr was about 1.8, which is close to the value of 2 of pristine monolayer Gr" is incorrect, see the work Casiraghi, C., Pisana, S., Novoselov, K. S., Geim, A. K., & Ferrari, A. C. (2007). Raman fingerprint of charged impurities in graphene. Applied physics letters, 91 (23), 233108.  I quote an essential passage of the abovementioned paper "We study more than 40 as-prepared monolayer graphenes, produced by microcleavage of graphite ............... The relative intensity of the 2D and G peaks strongly varies. ……. Note that Fig. 1 does not mean that the Raman spectra always vary in different samples or that they always change within a given sample. However, it warns that uniformity has to be checked, and cannot be simply assumed. Moreover, Fig. 1 dismisses the suggestion of Refs. 21 and 22 that either G peak position or I2D / IG can be used to estimate the number of layers, since the variation of these parameters in as deposited single layers far exceeds that assigned to the increase of number of layers. The criterium based on the shape of the 2D peak (ref.7) still stands and allows layer counting. "

Therefore the only valid criterion for evaluating the number of layers in graphene is the shape of the 2D peak.

Reviewer 3 Report

This manuscript entitled “Surface Properties of CVD-grown Graphene Transferred by

Wet and Dry Transfer Processes” by Min-Ah Yoon et al., have described the surface properties of transferred graphene via wet vs. dry transfer method. Such study must helpful for researchers working in different field of graphene applications. It can be publishable on Sensors after a major revision after addressing following important points:

  1. Authors have repeatedly emphasized the importance of such study for gas sensors, but have not presented any study related to the gas sensing activity of such wet and dry transferred graphene film. Inclusion of such study will be a proof of concept proposed by authors.
  2. Language need to be improved. The expressions are not clear in several lines, for instance: Page 1 “Gas sensors mainly utilize metal oxide films as the sensing material to precisely detect gas leakage at temperatures above a few hundred degrees Celsius [2-3]. The sensor’s problems are that the high operating temperature of gas sensors with metal oxide films can trigger an explosion of gases, such as hydrogen and oxygen, and its limited flexibility makes unsuitable for curved attachments on gas pipes and vessels; Although various transfer processes of graphene, such as wet, dry, mechanical, electrochemical, and polymer-free transfer, have been suggested, wet and dry transfer methods have been widely utilized because they are simple and produce high-quality graphene etc…..
  3. Page 2: “. In addition, polymeric residue on the graphene surface of gas sensors acts as a high-sensitivity functionalization layer for analyte adsorption [14]. However, this type of damage reduces the charge carrier mobility of graphene, thereby degrading the performance of the gas sensor; also, the residue increases the binding energy with gas molecules, resulting in gas sensor recovery issues [18]”, In this statement polymeric residue has been introduced without any prior details. Kindly improve this paragraph.
  4. Figure 2 Sheet resistance shows non uniformlity for dry transferred method, however the AFM images exhibits lower RMS roughness compared to wet. Authors should explain it in details.
  5. Page 9: conclusion: “After the transfer process, we observed little to no mechanical damage, such as cracks or wrinkles in Gr, regardless of the transfer method” This statement is contradicting the results presented in this article. Kindly improve this statement.
  6. Typos should be corrected: Line 80: 207 /Sq, Line 135: 5 × 5 μm2, Line 136: SiO2/Si etc…

Round 2

Reviewer 1 Report

I think the introduction section should have been improved as I suggested earlier. Moreover, I still believe that the manuscript does not fit into the sensor scope as nothing has been demonstrated related to sensing studies.

Reviewer 2 Report

Based on the answers received from the authors, I realize that I have not clearly expressed my thoughts and therefore I am not satisfied with their answers.

A work such as the one the authors intend to publish, which presents a comparison between two transfer techniques and which purports to provide general indications, must be rigorous in its statistical treatment. I understand well that the system created by the authors can minimize the randomness of the process, however there are several articles in the literature that demonstrate that the chemical-physical properties of the material can depend a lot on the interaction with the substrate and be different from point to point. (Again, see for example the work Casiraghi, C., Pisana, S., Novoselov, K. S., Geim, A. K., & Ferrari, A. C. (2007). Raman fingerprint of charged impurities in graphene. Applied physics letters, 91 (23), 233108), not to mention what happens when passing from one sample to another. I reiterate my indication: a statistical analysis must be carried out and I do not think 5 samples are enough.

Also, my suggestion about reinforcing a work weakness, which is a very mild link with sensor application, was completely ignored.

As regards the answer to point 2, the modification to the text proposed by the authors not only makes the concept even less clear than before, but is even expressed incorrectly; probably this is due to the fact that the authors make some confusion between the effect on the basic conductance of the material due to the presence of structural defects and the variation in conductance generated by the interaction with the analytes, thus confirming my feeling that they do not have much experience in the field of sensors and this also explains the very weak connection of working with this application. It was enough to simply change as follows: " Linear wrinkles and cracks in graphene restrict the path of the electrons, thus increasing the electrical resistance of graphene ", all the rest of the sentence must be deleted as incorrect.

Regarding the answer to point 3, I simply asked to specify that the side of the TF put in contact with graphene is the one where the PDMS is present, because this information cannot be obtained either in the text or in the figure.

That said, I find it pointless for the authors to re-submit the work to my review without the changes I have indicated

Reviewer 3 Report

Improvement has been done in the revised manuscript and can be accepted for publication.

Round 3

Reviewer 1 Report

The manuscript is in better shape now. It can be accepted for publication.

Reviewer 2 Report

Although the link with the application to the sensors is always quite weak, the authors have accepted all the indications I have provided, therefore I believe that  the work can be published in present form.